# Comparative Clinical Characteristics, Laboratory Findings, and Outcomes of Hypoxemic and Non-Hypoxemic Patients Treated at a Makeshift COVID-19 Unit in Bangladesh: A Retrospective Chart Analysis

**DOI:** 10.3390/jcm11112968

**Published:** 2022-05-24

**Authors:** Monira Sarmin, Mustafa Mahfuz, Lubaba Shahrin, Nusrat Jahan Shaly, Shamsun Nahar Shaima, Shamima Sharmin Shikha, Didarul Haque Jeorge, Shoeb Bin Islam, Mohammod Jobayer Chisti, Tahmeed Ahmed

**Affiliations:** International Centre for Diarrhoeal Disease Research, (icddr,b), Dhaka 1000, Bangladesh; drmonira@icddrb.org (M.S.); lubaba.shahrin@icddrb.org (L.S.); nusrat.jahan@icddrb.org (N.J.S.); shamsun.shaima@icddrb.org (S.N.S.); shamima.sharmin@icddrb.org (S.S.S.); didarul.haque@icddrb.org (D.H.J.); shoeb@icddrb.org (S.B.I.); tahmeed@icddrb.org (T.A.)

**Keywords:** Bangladesh, COVID-19, CRP, dyspnea, lymphopenia, makeshift hospital, older age

## Abstract

Background: Starting on 31 December 2019, from Wuhan City, China, Coronavirus disease 2019 (COVID-19) caused a global pandemic by 11 March 2020. Bangladesh detected its first case on 8 March 2020, only 66 days later the detection of the first case in China. We aimed to describe the epidemiology, clinical features, laboratory characteristics, and outcomes of Bangladeshi COVID-19 patients. Methods: This retrospective chart analysis compared Bangladeshi COVID-19 patients with hypoxemia compared to those without hypoxemia treated in a makeshift COVID-19 unit of icddr,b. Results: By March 2021, 207 remained in-patient. Nineteen patients (9.2%) died, whereas 10 (4.8%) were referred to different facilities for definitive care. Out of 207 in-patients, 88 patients required oxygen therapy. Multivariable logistic regression identified age (1.07 (1.02–1.13)), dyspnea (3.56 (1.06–11.96)), high CRP (1.13 (1.03–1.25)), and lymphopenia (6.18 (1.81–21.10)) as the independent predictors for hypoxemia in patients hospitalized for COVID 19 (for all, *p* < 0.05). Conclusion: Older age, dyspnea, high CRP, and lymphopenia are simple, but important, clinical and laboratory parameters. These may help clinicians to identify COVID-19 patients early who are at risk of fatal hypoxemia. Close monitoring, and prompt and aggressive treatment of these patients would curb their morbidity and mortality, especially in resource-limited settings.

## 1. Introduction

Starting from Wuhan City, China, 2019-nCoV caused by severe acute respiratory syndrome coronavirus 2 (SARS-CoV-2), later known as Coronavirus disease 2019 (COVID-19), affects millions of people throughout the world. WHO declared COVID-19 a global pandemic on 11 March 2020 [1]. As of 4 May 2022, COVID-19 infected over 512.6 million individuals worldwide, and has resulted in over 6.2 million deaths. Two-hundred and twenty-six countries have reported laboratory-confirmed cases of COVID-19 [2]. Bangladesh detected its first case on 8 March 2020, only 66 days after the first cases detected in China on 31 December 2019 [3]. Following exposure, symptoms of fever, cough, shortness of breath (SOB), myalgia, headache, sore throat, nausea, vomiting or diarrhea, and loss of taste or smell [4] develop between 2 days to 2 weeks. COVID-19 has a varied presentation, from mild symptoms to severe illness leading to fatality, and many patients remain asymptomatic throughout the disease course [4]. Transmission occurs mainly through close contact via respiratory droplets [5,6]. As a developing country with a GDP (per capita) of 2064 USD, Bangladesh is also struggling with its limited resources. Different government and non-government organizations came forward to fight against COVID-19. By April 2021, Bangladesh faced 11,450 deaths, with 7 million COVID-19-infected cases [3]. The International Centre for Diarrhoeal Disease Research, Bangladesh (icddr,b), one of the reputed research organizations in Bangladesh, also felt the need to fight against COVID-19 by providing clinical services to COVID-19-affected patients. In its premises, icddr,b Dhaka hospital set up a makeshift hospital to fight against COVID-19 in March 2020. They developed a management guideline based on national and international guidelines [5,6], and trained their staff on infection prevention and control, and clinical case management. The aim of this study was to describe the epidemiology, clinical features, laboratory characteristics, and clinical outcomes of all PCR-confirmed COVID-19 Bangladeshi patients managed in the icddr,b makeshift COVID-19 unit.

## 2. Materials and Methods

Study design: This is a retrospective chart analysis of all COVID-19 cases managed between April 2020 to March 2021 in a makeshift unit of icddr,b Dhaka hospital dedicated to COVID-19. 

Study site: icddr,b Dhaka hospital is the largest diarrheal hospital in the world. Here, about 150,000 patients are managed each year. At the beginning of the COVID-19 crisis in Bangladesh, they came forward and established a makeshift hospital to fight against COVID-19 in March 2020. A well-organized triage, an isolation unit, and 10 bedded COVID tents were prepared for patient care. This COVID-19 unit has facilities for intensive monitoring (cardiac and respiratory support) of patients with three high-flow nasal cannula (HFNC), two mechanical ventilators (MV), and five cardiac monitors with adequate syringe-pumps and infusion pumps. A dedicated management team was formed by physicians, nurses, and health assistants trained in critical medicine, and headed by an intensivist physician and scientist, and led by the senior director of the division. The team reviewed each case daily, and updated the management plan accordingly.

### 2.1. Operational Definition

(A)Clinical category of COVID-19 [5,6]

Mild case: Patient having an influenza-like illness with mild symptoms, for example, fever, cough, malaise, headache, sore throat, muscle pain without dyspnea, or abnormal imaging.

Moderate case: Adolescent or adult with fever, cough, dyspnea, RR < 30 breaths, and saturation > 93% without any oxygen support.

Severe cases: Cases with respiratory distress and RR ≥ 30 breaths/min or oxygen saturation ≤ 93% at rest. For the study purpose, cases having an acute myocardial infarction, COVID-19-associated coagulopathy, unremitting fever, or sepsis were also included in severe cases.

Critical cases: Patients requiring mechanical ventilation or HFNC for respiratory failure or having septic shock (hypotension that persists even after adequate fluid resuscitation, and requires vasopressors to sustain mean arterial pressure (MAP) ≥ 60 mmHg) or organ failure.

(B)Acute Respiratory Distress Syndrome (ARDS)

Acutely occurring respiratory failure not fully explained by cardiac failure or fluid overload having bilateral opacities, on chest imaging. In the absence of arterial blood gas analysis to evaluate hypoxemia, SpO_2_/FiO_2_ ≤ 315 is ARDS [6].

(C)Heart failure

In a dyspneic patient, tachycardia, the presence of dependent edema, and suggestive radiological changes with high BNP/NT-pro BNP. Due to extensive PPE use, we were unable to auscultate the lungs for the evaluation of basal crackles, which is an important finding of heart failure.

Case management: Following national [5] and international guidelines [6,7] icddr,b developed its clinical protocol that was revised over time to accommodate the new evidence-based recommendations. At triage of the makeshift unit, patients were screened, and all suspected patients were forwarded to a separate tent where they waited until COVID-19 testing was done. All diagnosed patients were categorized, and mild COVID-19 patients stayed in the isolation unit when they lacked the facilities for home isolation. Moderate, severe, and critical cases were managed in the COVID-19 tent, where almost all advanced medical facilities were available. Patients or their relatives measured oxygen saturation at home by portable pulse oximeter, and were admitted to the hospital if saturation dropped below 90% (94% during the initial days of COVID-19), or if symptoms worsened.

For them, complete blood count, coagulation profile, serum biochemical test (including renal and electrolytes), and chest X-ray (P/A for ambulant patients, and A/P for portable one) were performed initially. Other tests were performed according to clinical necessity. Famotidine, vit C, vit D, and Zinc were empirically administered. At the beginning of March and April 2020, patients were given hydroxychloroquine and azithromycin, and later we changed it to ivermectin (12 mg single dose) and doxycycline (100 mg 12 hourly for five days) from June 2020, and finally, from November 2020, only ivermectin (12 mg daily for 5 days) was prescribed based on the local evidence [8]. However, later studies found substantial unpredictability on the efficacy of ivermectin in COVID-19-affected patients [9].

Antibiotics were prescribed for patients who had a high level of C-reactive protein (CRP) with evidence of infection in complete blood count (CBC) or chest X-rays. According to the recovery trial, corticosteroid therapy was given at a dose of dexamethasone 6 mg daily for 10 days [10] for severe to critical cases. However, methylprednisolone was used in few cases before this evidence was surfaced. Oxygen therapy was administered to the patients using a nasal cannula, face mask, non-rebreather mask, HFNC, or invasive MV according to the severity of hypoxemia. HFNC was introduced in July 2020, and before that, patients with respiratory failure were managed with a non-rebreather mask with an oxygen flow rate of up to 15 L/min. Non-responsive patients were switched to an MV, or they were referred to other facilities.

Data collection: We reviewed clinical and nursing records, laboratory findings, and chest radiographs to get the required patient data between 1 April 2020 to 31 March 2021. We collected data on demographic and socioeconomic information (age, sex, education, occupation), presenting complaints such as fever, cough, sore throat, dyspnea, diarrhea, body ache, and generalized weakness; co-morbid conditions, such as hypertension, diabetes mellitus, chronic respiratory illness (bronchial asthma or COPD), and ischemic heart diseases (H/O coronary intervention such as coronary artery bypass grafting (CABG) or angiogram, angina); as well as vital signs (pulse, blood pressure (BP), temperature, respiratory rate (RR), arterial oxygen saturation (SpO_2_), random blood sugar (RBS)), physical examination findings (pallor, cyanosis, breathing pattern, edema, dehydration, organomegaly), the requirement of oxygen support (by nasal cannula or NRM or HFNC or MV) laboratory reports on complete blood count (CBC), electrolyte and creatinine, coagulation profile (D-dimer), inflammatory marker (C-reactive protein (CRP)), and chest x-ray. Moreover, we collected data on treatment outcomes that included discharge, referral, or death or complications that developed during the disease course, such as acute respiratory distress syndrome (ARDS), septic shock, and heart failure. A research assistant entered the data, and four physicians of the COVID-19 management unit reviewed it. Depending on clinical conditions, different laboratory tests were performed at different time points. To maintain homogeneity, we only included the baseline (performed within 2 days before or after admission) investigations in the analysis.

### 2.2. Statistical Analysis

Quantitative variables were presented as median (IQR) for heterogenous data, or mean (±SD) for homogenous data, and compared with the Mann–Whitney U test and *t*-test, respectively. Qualitative variables were presented as frequency. χ^2^ test or Fisher’s exact test compared their significance as appropriate. A two-sided probability of <0.05 was considered statistically significant. We reported OR with their 95% CI to represent the strength of association. Finally, a multivariable logistic regression analysis (enter method) was employed to identify independent variables that were significantly associated with the outcome of oxygen requirement. SPSS software (version 20) and Epi Info (version 7.0) were used for the analysis. Information Technology Department of icddr,b provided the software.

## 3. Results

Within 12 months of the study period, a total of 207 RT-PCR-confirmed COVID-19 cases received management from this inpatient facility. Twelve of them had an age <18 years and experienced an uneventful recovery. Of the 207 inpatients, 88 (42.5%) were hypoxemic and required intensive support. Among the hypoxemic adults, 54 improved solely by low flow oxygen administered through a nasal cannula (up to 6 L/min), three were referred (acute myocardial infarction (1), cerebrovascular accidents/CNS infection (1), attendant’s preference (1)), and the remaining 31 required an escalation of oxygen therapy using a face mask (FM). Among the FM group, 29 required an escalation of oxygen by using NRM (up to 15 L/min) due to the worsening of hypoxemia, and one required MV. Out of 29, 15 received HFNC, and four required MV. Three were referred (unavailability of HFNC (3), and attendant’s preference (1)). Up to July 2020, HFNCs were not available at our treatment facility. We have started HFNC with a flow of 20 L/min. Two patients who received less than 40 L/min of O_2_ by HNFC were weaned off after 5 days and 9 days, respectively. However, we escalated up to 80 L/min for others before introducing them to MV or reaching the fatal outcome. Two patients recovered and got discharged, one deescalated to NRM and was referred (for complete heart block), five patients were intubated for mechanical ventilation, and due to refusal from the patients’ attendant, seven patients were not intubated. All patients in MV (*n* = 10) did not survive. Out of 119 non-hypoxemic patients, all were discharged except three (referred due to attendant’s preference (2), and renal calculi (1)) (Figure 1).

COVID-19 in-patient characteristics: Patients who required oxygen were older than those who did not require oxygen (median age, 58 y vs. 42 y), and they also had more comorbidities such as hypertension, diabetes, and IHD. They presented with shortness of breath, fever, and tachypnea at admission (*p*-value for all, <0.05). Other baseline features were comparable between the groups (Table 1).

Our data showed that patients who required oxygen therapy during hospitalization more often had WBC aberration with leucopenia or leukocytosis (white blood cell count less than 4 × 10⁹/L or greater than 11 × 10⁹/L, respectively), lymphopenia, higher D-Dimer and CRP, hyponatremia, and metabolic acidosis on admission compared to those who did not require oxygen therapy (Table 2).

On admission, a chest x-ray was performed in 140 cases where bilateral X-ray changes were observed among 67.5% of patients in the severe-critical group, and 14% of patients in the mild-moderate group (Table 2).

There were 19 deaths, and all of them were from the oxygen-required group. Those who required oxygen therapy also developed several complications during the disease course, e.g., ARDS, heart failure, and septic shock (Figure 2a,b). Compared to patients who did not require oxygen, they required more injectable antibiotics, and, more often, their antibiotics required an escalation to a higher generation. They also consumed more insulin, proton pump inhibitors, and steroids (Appendix A, Appendix A). As of March 2021, 86% of the 207 admitted patients recovered and had been discharged successfully.

Multivariable logistic regression (enter method) was performed with the clinically significant variables from bivariate analysis to find independent factors predicting oxygen therapy among the COVID-19-infected patients. After adjusting for potential confounders, older age, dyspnea, high CRP, and lymphocytopenia were found to predict oxygen therapy among the COVID-19-infected patients (Table 3).

The receiver operating characteristic (ROC) curve with area under the curve (AUC) describes the contribution of CRP, lymphopenia, dyspnea, and age for hypoxemia among the patients hospitalized for COVID-19 (Figure 3).

## 4. Discussion

To our knowledge, this is the first reported study from Bangladesh where a make-shift hospital managed COVID-19 infected patients with intensive monitoring and follow-ups. We found older age, dyspnea/shortness of breath, and laboratory characteristics such as high CRP and lymphocytopenia (low lymphocyte) at admission were associated with oxygen requirement among COVID-19-infected patients. Here, the case fatality rate was 9.2% (19/207) among the moderate to critical cases. It varied in different regions of the world, and was found to have ranged from 3.14 to 3.5% among the outpatient cases [11,12]. A systematic review showed that the pooled prevalence of mortality among hospitalized patients with COVID-19 was 17.62% (95% CI, 14.26–21.57%) [13]. The most important observation of this study is the low mortality compared to other makeshift COVID centers treating severe and critically ill COVID-19 patients [14,15]. Perhaps, proper patient care, protocolized management based on WHO and national guidelines, early adoption of evidence-based new treatment modalities during COVID-19, regular refreshers training for all the health care providers, continuous monitoring and supervision by specialists’ doctors, daily group discussions, and, most importantly, a team of dedicated doctors and nurses are the factors that resulted in the higher survival of our patients. Besides that, external consultancy from specialist physicians was sought when we found it necessary.

Older age was found to be a strong predictor of in-patient morbidity and mortality of patients admitted with COVID-19 and is included in other existing COVID-19 severity scores also [16,17,18,19]. Age is always an important factor in the clinical course of a disease. Extreme age was identified as an important predictor of mortality in severe acute respiratory syndrome (SARS) and Middle East Respiratory Syndrome (MERS) [20,21]. As people get older, there is a dysregulation of the immune system, particularly due to the development of an adaptive immune response with preserved innate immunity [22]. There is a loss of CD4 cells, shift from Th1 to Th2 cytokines with persistent hyper inflammation, expansion of cytotoxic CD8 cells, and minimum functional impairment of Neutrophil, monocyte/macrophage, and natural killer cells. Thus, the collective impact of all these defects in aged persons results in a reduced ability to respond against a new pathogen in an effective way [23,24]. A recent meta-analysis by Parohan et al. revealed that extreme age is an important risk factor for death among patients affected with COVID-19 [18]. In our cohort, the observation of a higher risk of disease progression with oxygen support among elderly patients is consistent with the above findings.

COVID-19-infected patients also presented with shortness of breath, indicating more pronounced involvement of lung parenchyma and subsequent development of hypoxemia. By increasing an upregulation of receptors from the afferent pulmonary C fibers, viral infection can cause shortness of breath [25] which was evident earlier in influenza (82%) and MERS (69%) [16,26,27], whereas in COVID-19, shortness of breath varied and may present quite differently, either having normal breathing with “silent” hypoxemia or remarkable dyspnea among different cohorts [28]. C-reactive protein (CRP), an acute-phase inflammatory protein is primarily synthesized and stored in hepatocytes, and released in response to infection and inflammation, primarily with the stimulation of IL-6 followed by IL-1 and tumor necrosis alpha (TNF-α) [29]. In COVID-19, several studies found that an increased CRP was associated with disease severity and poor progression [30,31]. A Swedish multicenter study disclosed that an admission CRP level > 10 mg/dL was used as the predictor of ICU admissions and 30-day mortality [32].

At the beginning of inflammation, CRP binds to pathogens, activates the complement pathway, and promotes phagocytosis and apoptosis. However, by inhibiting neutrophil chemotaxis and delaying apoptosis, CRP expresses its anti-inflammatory effects also [33]. In COVID-19, SARS-CoV-2 damages organs by inflammatory response, resulting in the progression of diseases [33]. Thus, markedly elevated CRP might act as a surrogate of excessive inflammation, leading to severe and critical illness, and even deaths in COVID-19. Studies from other centers also echoed the same findings of higher D-dimer and CRP in predicting severe COVID-19 infections from that of mild-moderate diseases [34,35].

Our data also depicted the same findings of higher CRP predicting the oxygen requirement of COVID-19-infected patients, thus differentiating severe cases from mild-moderate cases. In our analyses, D-dimer was significantly higher among the COVID-19 patients requiring oxygen therapy than those who did not require oxygen therapy in bivariate analysis; however, the significance was lost in multivariable analysis.

Among the hematological parameters, leukocytosis, lymphopenia, and increased neutrophil to lymphocyte ratio were associated with disease severity. A recent review regarding COVID-19 by Jafarzadeh et al. found the frequency of lymphopenia was 0.6–80.4% and 32.7–96.1% among mild/moderate and severe COVID-19 cases, respectively [36]. Lymphopenic patients had a three-fold higher risk of developing severe COVID-19 [37], and survived less than non-lymphopenic patients, indicating that the repletion of lymphocytes may impact recovery [38]. In SARS, the reported prevalence of lymphopenia was 69.6–54% [39]. SARS-CoV-2 may also cause lymphocytopenia by direct infection, pyroptosis, bone marrow impairment, thymus suppression, tissue redistribution, cytokine and antibody-mediated destruction, and autophagy [36,37]. In our studied COVID-19 cohort, lymphocytopenia was observed to be associated with 6.18 times higher odds of disease severity and oxygen requirement.

We had several limitations. Being a retrospective and single center study, all relevant data were not available for all patients, and we might have missed some unknown risk factors for the requirement of oxygen therapy. However, as we know, this is the first study from Bangladesh; hence, data may be important for other clinicians and policymakers in limited resources settings such as Bangladesh.

## 5. Conclusions

Our data analyses demonstrate that older age, dyspnea, high CRP, and lymphopenia are associated with an oxygen requirement among the COVID-19-affected hospitalized patients. We need to remain vigilant for COVID-19-affected patients having these clinical and laboratory characteristics. Thus, our observation of clinical and laboratory characteristics of COVID-19 patients and their predicting factors for developing hypoxemia are found to be consistent among the millions of COVID-19-affected people globally.

## Figures and Tables

**Figure 1 jcm-11-02968-f001:**
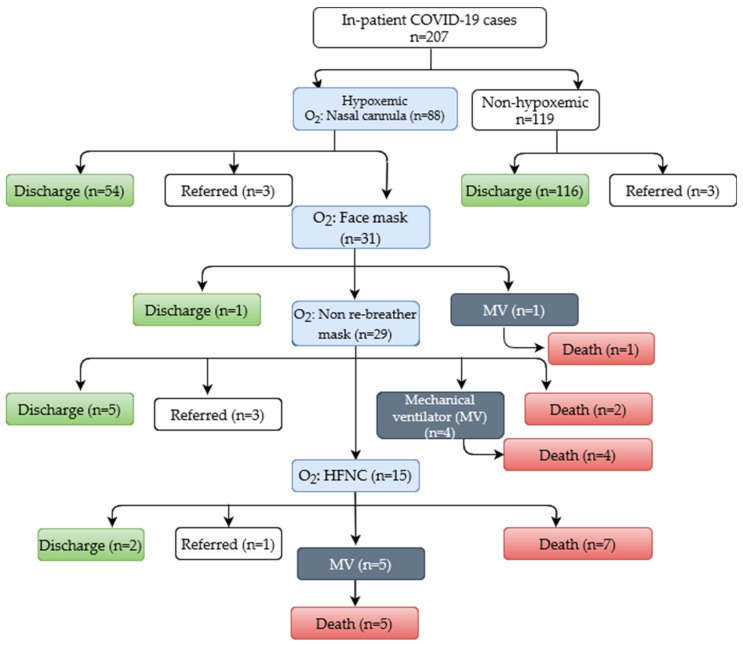
Flow diagram showing COVID-19 patients managed at icddr,b Dhaka Hospital.

**Figure 2 jcm-11-02968-f002:**
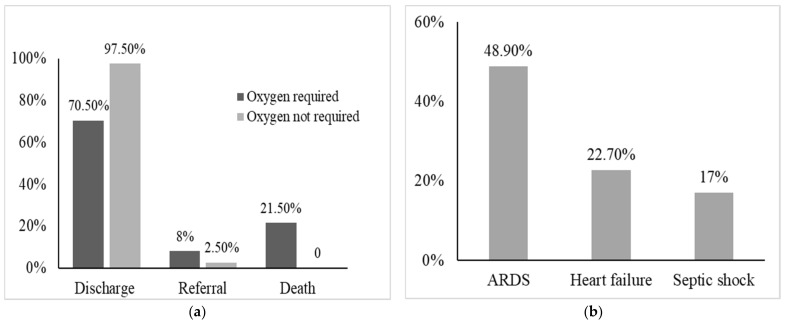
(**a**) Outcome of the study patients, and (**b**) complications during the disease course among the patients requiring oxygen (ARDS = acute respiratory distress syndrome).

**Figure 3 jcm-11-02968-f003:**
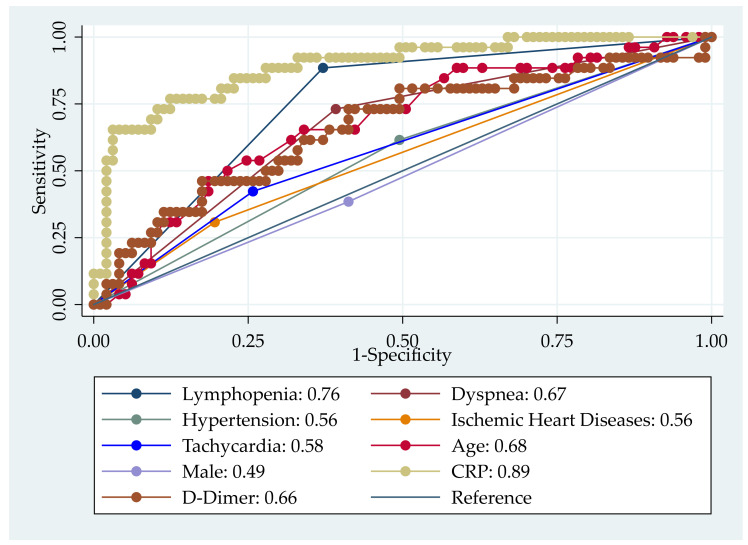
Receiver operating characteristic (ROC) curve with area under the curve (AUC) showing parameters associated with hypoxemia among the patients hospitalized for COVID-19.

**Table 1 jcm-11-02968-t001:** Demographics and baseline characteristics of in-patient COVID-19 cases according to oxygen requirement.

Characteristics	Oxygen Required (*n* = 88)	Oxygen Not Required (*n* = 119)	OR, 95%CI	*p* Value
Age (years) (median, IQR)	58 (51.2, 65)	42 (35, 54)	-	<0.001
Sex, male	54 (61.4)	74 (62.7)	0.94 (0.53–1.66)	0.95
Days from illness onset to admission (median, IQR)	7 (4, 8)	5 (3, 9.5)	-	0.54
Antibiotics before hospitalization	18 (20.5)	18 (15.1)	1.44 (0.70–2.97)	0.415
Presence of comorbidity				
Hypertension	54 (61.4)	40 (33.6)	3.13 (1.75–5.59)	<0.001
Diabetes Mellitus	41 (46.6)	32 (26.9)	2.37 (1.32–4.25)	0.005
Ischemic Heart Disease	26 (29.9)	7 (5.9)	6.71 (2.75–16.34)	<0.001
Chronic respiratory illness (asthma and COPD)	15 (17.4)	18 (15.5)	1.15 (0.54–2.43)	0.86
Hypothyroidism	8 (9.1)	8 (6.7)	1.39 (0.50–3.85)	0.713
Signs and symptoms				
Fever	70 (80.5)	82 (68.9)	1.85 (0.96–3)	0.08
Cough	57 (66.3)	76 (64.4)	0.81 (0.46–1.41)	0.54
Sore throat	5 (5.8)	13 (11.1)	0.49 (0.16–1.44)	0.28
Headache	13 (15.1)	25 (21.4)	0.65 (0.31–1.36)	0.34
Diarrhea	10 (11.6)	12 (10.3)	1.15 (0.47–2.80)	0.93
Shortness of breath (SOB)	48 (54.5)	33 (27.7)	3.13 (1.75–5.59)	<0.001
Body ache/myalgia	8 (9.4)	19 (16.2)	0.53 (0.22–1.28)	0.23
Systolic blood pressure				
Normotensive, <120 mm (ref)	30 (36.6)	47 (42.3)	-	-
Pre-hypertensive (120–139) mm of hg	31 (37.8)	44 (39.6)	1.10 (0.58–2.11)	0.894
Hypertension (>140 mm of hg)	21 (25.6)	20 (18)	1.64 (0.76–3.53)	0.278
Diastolic Blood Pressure				
Normotensive <80 mm of hg (ref)	48 (58.5)	46 (41.4)	-	-
Pre-hypertensive (80–89) mm of hg	21 (25.6)	37 (33.3)	0.54 (0.28–1.06)	0.105
Hypertension >90 mm of hg	13 (15.9)	28 (25.2)	0.44 (0.20–0.96)	0.059
Temperature >38 °C	17 (21.2)	8 (7.1)	3.54 (1.44–8.68)	0.007
RBS (median, IQR)	8 (6.3,10.7)	7.35 (5.8,10.9)	-	0.414
Tachycardia (Heart rate > 100/min)	27 (32.5)	20 (17.7)	0.44 (0.22–0.86)	0.025
Tachypnoea (Respiratory rate > 30/min)	39 (50)	10 (9.1)	10 (4.55–21.9)	<0.001

**Table 2 jcm-11-02968-t002:** Laboratory profiles of in-patient COVID-19 cases at admission.

Characteristics	Patients Required Oxygen (*n* = 88)	Patients Required No Oxygen (*n* = 119)	OR, 95% CI	*p* Value
Complete Blood Count				
Hb (g/dL)	12.2 ± 1.6	12.5 ± 1.6	-	0.191
Leukocytosis or leucopenia	23 (28)	9 (11.7)	2.94 (1.26–6.86)	0.018
Lymphopenia	54 (65.9)	18 (23.4)	6.32 (3.15–12.7)	0.000
Platelet (×10^9^/L)	211.35 ± 89.8	229.1± 91.1	-	0.218
CRP (mg/dL)	9.4 (3.0, 19.9)	1.3 (0.3, 4.5)	-	0.001
D-Dimer (pg/mL)	659 (360.5, 1028.5)	409.5 (277.1, 673.7)	-	0.010
Serum electrolyte				
Hyponatremia	23 (31.9)	8 (17.4)	2.23 (0.9–5.53)	0.124
Hypokalemia	18 (25.4)	16 (34.8)	0.64 (0.28–1.43)	0.374
Metabolic acidosis	39 (54.9)	9 (19.6)	5.01 (2.11–11.91)	<0.001
Normal renal function	59 (90.8)	37 (100)	-	0.142
Bilateral involvement in chest X-rays	52 (67.5)	9 (14.3)	12.48 (5.32–29.25)	<0.001

Leukocytosis or leucopenia (WBC count < 4 × 10^9^/L or >11 × 10^9^/L); lymphopenia (absolute lymphocyte count < 1.5 × 10^9^/L); hyponatremia (serum sodium < 135 m mol/L); hypokalemia (serum K < 3.5 m mol/L); metabolic acidosis (TCO_2_ < 23 m mol/L).

**Table 3 jcm-11-02968-t003:** Factors associated with oxygen requirement in admitted COVID-19-infected patients.

Characteristics	OR, 95% CI	*p* Value
Age	1.07 (1.02–1.13)	0.005
Sex male	0.54 (0.16–1.85)	0.326
Hypertension	1.82 (0.56–5.92)	0.319
Ischemic Heart Diseases	3.83 (0.81–18.24)	0.091
Diabetes Mellitus	0.59 (0.15–2.24)	0.437
Dyspnea	3.56 (1.06–11.96)	0.040
Fever	1.16 (0.24–5.68)	0.857
Tachycardia	1.48 (0.44–4.93)	0.522
Lymphopenia	6.18 (1.81–21.10)	0.004
CRP	1.13 (1.03–1.25)	0.011
D-Dimer	1.0 (0.99–1.00)	0.382

## Data Availability

The Institutional Review Board, icddr,b, has the right to share data upon request. Data requests may be sent to Armana Ahmed (aahmed@icddrb.org), Head, Research Administration.

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
