# Peer review of "Comparative Clinical Characteristics, Laboratory Findings, and Outcomes of Hypoxemic and Non-Hypoxemic Patients Treated at a Makeshift COVID-19 Unit in Bangladesh: A Retrospective Chart Analysis"

_jcm, 2022, doi:10.3390/jcm11112968_

Round 1

Reviewer 1 Report

It was a pleasure to read the excellently written article. The manuscript is written flawlessly. Almost no typo can be found (Ln 139 2days; table 3 herat disease, Ln 213 Hete). English is outstanding.
The introduction is concise, the methods are described and clarified in the smallest detail. I especially commend the definitions of the observed clinical conditions (mild/severe disease, ards…), the detailed description of the patient care protocol and the escalation of oxygen therapy, and the adaptation to new phenomena and guidelines.
The results section is written correctly. From baseline characteristics, finding variables that could affect the necessity of oxygen therapy and, finally, multivariate analysis and defining clinical and laboratory entities that can predict the necessity of oxygen therapy. Here are my first objections:
1. "refusal from caregivers" - I'm not an English speaker, but caregiver refers too much to a healthcare provider, and this is obviously about people who are close to these patients, so consider rephrasing to avoid misunderstanding
2. Figure 1 - excellent diagram, please correct n = 01 to n = 1 in similar instances
3. I must point out an excellent representation of data in tables. Just a positive remark.
4. Was enter or another (preferably stepwise method) used in logistic multivariate regression? Please add to the statistical part.
5. I advise doing an equation with the coefficients resulting from multivariate analysis and then showing the obtained score for each case as an HSROC curve according to the use of oxygen therapy or MV or death. This would probably contribute to a stronger presentation of the results.
The discussion is wide, I have no significant objections here if it fits into the numerical size provided in journal policies. Conclusions follow the results, and the generalization is good.
But unfortunately, despite the excellent set and methodological presentation, the result is self-evident.
Dyspnoea without covid19 infection is an indication for oxygen therapy in most conditions, age is a significant predictive factor for death or escalation of oxygen therapy described in many studies. The same applies to CRP and lymphopenia, which are biochemical parameters previously associated with the prediction of poor outcomes. And as the authors stated, the limit is a retrospective from the time since the beginning of the pandemic with a relatively small number of cases. I still think that this is a good study whose results should be published, but more to fit into some synthesis of evidence - systematic review and meta-analysis, because at the moment they do not bring an essential novelty. I agree that this is the first study from Bangladesh, and hence data may be important for other clinicians and policymakers in limited-resource settings.

Author Response

Rebuttal letter

Date:   May 8, 2022

To:   Aileen Liu
        Editor
       Journal of Clinical Medicine

From:    Dr Md Jobayer Chisti & Dr Mustafa Mahfuz

             Corresponding authors,

             Manuscript ID jcm-1666780

Subject:   Response on the comments of the Reviewers of JCM Manuscript ID jcm-1666780 entitled “Comparative clinical characteristics, laboratory findings, and outcomes of hypoxemic and non-hypoxemic patients treated at a make-shift COVID-19 unit in Bangladesh: a retrospective chart analysis”

Dear Aileen Liu,

Thank you for providing us the opportunity to resubmit our manuscript following revision. We greatly appreciate helpful and precise comments from the respected Reviewers and we have attempted to address them. We now submit the revised manuscript, which highlights the changes in track changes, as well as this cover letter which provides a point-by-point response to the reviewers’ comments.

Thank you

Response to reviewer’s comment

Reviewer`s one:

Comments and Suggestions for Authors

It was a pleasure to read the excellently written article. The manuscript is written flawlessly. Almost no typo can be found (Ln 139 2days; table 3 herat disease, Ln 213 Hete). English is outstanding.
The introduction is concise, the methods are described and clarified in the smallest detail. I especially commend the definitions of the observed clinical conditions (mild/severe disease, ards…), the detailed description of the patient care protocol and the escalation of oxygen therapy, and the adaptation to new phenomena and guidelines.
The results section is written correctly. From baseline characteristics, finding variables that could affect the necessity of oxygen therapy and, finally, multivariate analysis and defining clinical and laboratory entities that can predict the necessity of oxygen therapy. Here are my first objections:
1. "refusal from caregivers" - I'm not an English speaker, but caregiver refers too much to a healthcare provider, and this is obviously about people who are close to these patients, so consider rephrasing to avoid misunderstanding

Response:  Thank you so much for your appreciation. Now we have corrected the typos. We have rephrased caregivers to patient’s attendant to avoid misunderstanding (Table 3, page 6)

  1. Figure 1 - excellent diagram, please correct n = 01 to n = 1 in similar instances

Response: Thank you for valuing our work. Now we have changed the n. (page 5 )

  1. I must point out an excellent representation of data in tables. Just a positive remark.

Response: We are grateful for the positive comments.

  1. Was enter or another (preferably stepwise method) used in logistic multivariate regression? Please add to the statistical part.

Response: You are right. We have added enter method. Now we have added it to the statistical part (page 4).

  1. I advise doing an equation with the coefficients resulting from multivariate analysis and then showing the obtained score for each case as an HSROC curve according to the use of oxygen therapy or MV or death. This would probably contribute to a stronger presentation of the results.

Response: Thank you. Following your suggestion, we have done ROC with AUC to find the contribution of different covariates for hypoxemia among patients hospitalized for COVID-19 (page 11).

The discussion is wide, I have no significant objections here if it fits into the numerical size provided in journal policies. Conclusions follow the results, and the generalization is good.

Response: Thank you for allowing us to provide a detailed description.

But unfortunately, despite the excellent set and methodological presentation, the result is self-evident.
Dyspnoea without covid19 infection is an indication for oxygen therapy in most conditions, age is a significant predictive factor for death or escalation of oxygen therapy described in many studies. The same applies to CRP and lymphopenia, which are biochemical parameters previously associated with the prediction of poor outcomes. And as the authors stated, the limit is a retrospective from the time since the beginning of the pandemic with a relatively small number of cases. I still think that this is a good study whose results should be published, but more to fit into some synthesis of evidence - systematic review and meta-analysis, because at the moment they do not bring an essential novelty. I agree that this is the first study from Bangladesh, and hence data may be important for other clinicians and policymakers in limited-resource settings.

Response: Thank you for your observation. Though our findings are not novel, these findings highlight that simple clinical and laboratory tests could guide us to identify people at risk for clinical deterioration.

Reviewer 2 Report

In this paper Sarmin et al. described the population treated for COVID-19 in a single center in Bangladesh. The authors retrospectively gathered the clinical information of 207 inpatients. The population of the study varied from mild cases to severe COVID-19 patients. The aim of the study was to evaluate the characteristic of the population treated. The main conclusions of the study are that older age, inflammation, shortness of breath and lymphopenia are associated to the need of oxygen supplementation.

This paper is surely clear and well written, with a complete presentation of study methods and clinical setting of the population. However. the results are not a novelty in the literature and are based on a retrospective study. Furthermore, there are some criticisms to the methodology and the conclusions that should be raised:

  • The authors made a multivariate analysis to evaluate independent predictors for oxygen therapy; however, is not clear if some of the patients already needed oxygen at admission (as probable due to the long history of symptoms before admission). In case of patients already needing oxygen therapy, is not possible to understand if some of the parameters are causes or consequences of the hypoxemia (especially dyspnea and CRP). The authors should evaluate if the patients already presented hypoxemia at the time of evaluation of this parameters, and remove the ones that could be biased by the hypoxemia from the multivariate analysis. A better outcome to evaluate should be the need of high flows of oxygen (non rebrethable masks, HFNC or MV) during the hospitalization.
  • The authors claimed that the mortality rate among their population is low. However, they compared their heterogeneous population (mild to critical patients), with papers including only ICU admitted or critical patients (references 10 and 11). In papers including more heterogeneous patients the mortality was lower (e.g. in J Med Virol. 2022 May;94(5):2222-2229. The mortality among the placebo cohort was 3.5%). This conclusion should be revaluated.
  • The authors claimed that the use of ivermectin was based on local evidence. The use in the early stages of the pandemic based on early literature is acceptable. However, metanalysis found that the use of ivermectin had no efficacy in COVID-19 patients (e.g. J Clin Epidemiol. 2022 Apr;144:43-55.). This point should be added to the paper to avoid the idea of efficacy of ivermectin.
  • The authors evaluated the Chest X-Ray of all the population, but only for bilateral opacities. There are some standardized chest x ray scores that could be used and added to the paper as predictors of level of care (e.g. J Clin Med. 2020 Sep 16;9(9):2990., Radiology. 2020 Oct;297(1):E197-E206., Eur Radiol. 2021 Apr;31(4):1999-2012.)
  • A minor point that should be amended is in the abstract, where the percentages about dead and referred patients are incorrect.
  • In the introduction the authors cite the data about mortality and infection wordwide at February 2021.The citations should be updated to the current situation.

Author Response

Rebuttal letter

Date:   May 8, 2022

To:   Aileen Liu
        Editor
       Journal of Clinical Medicine

From:    Dr Md Jobayer Chisti & Dr Mustafa Mahfuz

             Corresponding authors,

             Manuscript ID jcm-1666780

Subject:   Response on the comments of the Reviewers of JCM Manuscript ID jcm-1666780 entitled “Comparative clinical characteristics, laboratory findings, and outcomes of hypoxemic and non-hypoxemic patients treated at a make-shift COVID-19 unit in Bangladesh: a retrospective chart analysis”

Dear Aileen Liu,

Thank you for providing us the opportunity to resubmit our manuscript following revision. We greatly appreciate helpful and precise comments from the respected Reviewers and we have attempted to address them. We now submit the revised manuscript, which highlights the changes in track changes, as well as this cover letter which provides a point-by-point response to the reviewers’ comments.

Thank you

Response to reviewer’s comment

Reviewers 2

Comments and Suggestions for Authors

In this paper Sarmin et al. described the population treated for COVID-19 in a single center in Bangladesh. The authors retrospectively gathered the clinical information of 207 inpatients. The population of the study varied from mild cases to severe COVID-19 patients. The aim of the study was to evaluate the characteristic of the population treated. The main conclusions of the study are that older age, inflammation, shortness of breath and lymphopenia are associated to the need of oxygen supplementation.

This paper is surely clear and well written, with a complete presentation of study methods and clinical setting of the population. However. the results are not a novelty in the literature and are based on a retrospective study. Furthermore, there are some criticisms to the methodology and the conclusions that should be raised:

  • The authors made a multivariate analysis to evaluate independent predictors for oxygen therapy; however, is not clear if some of the patients already needed oxygen at admission (as probable due to the long history of symptoms before admission). In case of patients already needing oxygen therapy, is not possible to understand if some of the parameters are causes or consequences of the hypoxemia (especially dyspnea and CRP). The authors should evaluate if the patients already presented hypoxemia at the time of evaluation of this parameters, and remove the ones that could be biased by the hypoxemia from the multivariate analysis. A better outcome to evaluate should be the need of high flows of oxygen (non rebrethable masks, HFNC or MV) during the hospitalization.

  • The authors claimed that the mortality rate among their population is low. However, they compared their heterogeneous population (mild to critical patients), with papers including only ICU admitted or critical patients (references 10 and 11). In papers including more heterogeneous patients the mortality was lower (e.g. in J Med Virol. 2022 May;94(5):2222-2229. The mortality among the placebo cohort was 3.5%). This conclusion should be revaluated.

Response: Thank you so much for suggesting the article. In this systematic review, for COVID-19 outpatient cases, trials were conducted between monoclonal antibodies and placebo. Among the placebo group, mortality was 3.5%. Another systematic review showed the pooled prevalence of mortality among hospitalized patients with COVID-19 was 17.62% (Dessie, Z.G., Zewotir, T. Mortality-related risk factors of COVID-19: a systematic review and meta-analysis of 42 studies and 423,117 patients. BMC Infect Dis 21, 855 (2021). https://doi.org/10.1186/s12879-021-06536-3). For this retrospective analysis, we excluded out-patient cases who had mild illnesses. Now we have addressed this in the discussion section (page 11).

  • The authors claimed that the use of ivermectin was based on local evidence. The use in the early stages of the pandemic based on early literature is acceptable. However, metanalysis found that the use of ivermectin had no efficacy in COVID-19 patients (e.g. J Clin Epidemiol. 2022 Apr;144:43-55.). This point should be added to the paper to avoid the idea of efficacy of ivermectin.

  • Response: Thank you. Now we have cited this paper and mentioned that ivermectin had a substantial unpredictability for COVID-19 (page 3).

  • The authors evaluated the Chest X-Ray of all the population, but only for bilateral opacities. There are some standardized chest x ray scores that could be used and added to the paper as predictors of level of care (e.g. J Clin Med. 2020 Sep 16;9(9):2990., Radiology. 2020 Oct;297(1):E197-E206., Eur Radiol. 2021 Apr;31(4):1999-2012.)

Response: Thank you for suggesting the papers. At this moment we have only information regarding the area of involvement- unilateral/bilateral/normal and an additional paper involving CXR evaluation is in progress, we are unable to use the score for predictions in this paper.

  • A minor point that should be amended is in the abstract, where the percentages for dead and referred patients are incorrect.

Response: We appreciate your observation. We have corrected the percentages in the abstract (page 1).

  • In the introduction, the authors cite the data about mortality and infection worldwide in February 2021. The citations should be updated to the current situation.

Response: Thank you for the suggestion. Now we have provided an update on the current situation. (Page 1).

Round 2

Reviewer 1 Report

Since all of my recommendations have been accepted and acknowledged I don't find any new suggestions for the authors. The only thing I could object to is using the enter method for the regression model (perhaps, the stepwise method would have excluded the CRP and/or age as predictors). But, properly stated this is acceptable. Thank you for accepting my advice on ROC presentation in your work.

Author Response

Response to Comments and Suggestions for Authors

Since all of my recommendations have been accepted and acknowledged I don't find any new suggestions for the authors. The only thing I could object to is using the enter method for the regression model (perhaps, the stepwise method would have excluded the CRP and/or age as predictors). But, properly stated this is acceptable. Thank you for accepting my advice on ROC presentation in your work.

Response: Thank you so much for your appreciation. Using the ROC curve was a nice suggestion.

Reviewer 2 Report

I thank the authors for the complete response to my points, as already stated by me and the other reviewer i think this is a really well written paper.

I belive that the part of the response given to my first point ("Patients or their relatives measured oxygen saturation at home by portable pulse oximeter and admitted to the hospital if saturation drops below 90% (94% during the initial days of covid), or if symptoms worsen.") should be added to your paper in the methods section to better clarify the study population.

Author Response

Response to Comments and Suggestions for Authors

I thank the authors for the complete response to my points, as already stated by me and the other reviewer i think this is a really well written paper.

I believe that the part of the response given to my first point ("Patients or their relatives measured oxygen saturation at home by portable pulse oximeter and admitted to the hospital if saturation drops below 90% (94% during the initial days of covid), or if symptoms worsen.") should be added to your paper in the methods section to better clarify the study population.

Response: Thank you so much for your appreciation. Now we have added the description “Patients or their relative measured oxygen saturation at home by portable pulse oximeter and admitted to the hospital if saturation drops below 90% (94% during the initial days of covid), or if symptoms worsen” in the method section following the recommendation from the respected reviewer (page 3).
